# A Comparative Genomic Approach to Determine the Virulence Factors and Horizontal Gene Transfer Events of Clinical Acanthamoeba Isolates

Xiaobin Gu,[b] Xiuhai Lu,[d,e,f] Shudan Lin,[b] Xinrui Shi,[a,c] Yue Shen,[a,c] Qingsong Lu,[a,c] Yiying Yang,[a,c] Jing Yang,[a,c] Jiabei Cai,[a,c] Chunyan Fu,[a,c] Yongliang Lou,[b] Meiqin Zheng[a,b,c]

[a]Eye Hospital and School of Ophthalmology and Optometry, Wenzhou Medical University, Wenzhou, Zhejiang, China

[b]Zhejiang Provincial Key Laboratory for Technology and Application of Model Organisms, Key Laboratory of Laboratory Medicine, Ministry of Education, School of Laboratory Medicine and Life Sciences, Wenzhou Medical University, Wenzhou, Zhejiang, China

[c]National Clinical Research Center for Ocular Diseases, Wenzhou, Zhejiang, China

[d]Eye Hospital of Shandong First Medical University (Shandong Eye Hospital), Shandong, China

[e]State Key Laboratory Cultivation Base, Shandong Provincial Key Laboratory of Ophthalmology, Shandong Eye Institute, Shandong First Medical University & Shandong Academy of Medical Sciences, Shandong, China

[f]School of Ophthalmology, Shandong First Medical University, Shandong, China

Xiaobin Gu and Xiuhai Lu contributed equally to this article. Author order was determined on the basis of contribution.

**ABSTRACT** *Acanthamoeba* species are among the most ubiquitous protists that are widespread in soil and water and act as both a replicative niche and vectors for dispersal. They are the most important human intracellular pathogens, causing *Acanthamoeba* keratitis (AK) and severely damaging the human cornea. The sympatric lifestyle within the host and amoeba-resisting microorganisms (ARMs) promotes horizontal gene transfer (HGT). However, the genomic diversity of only *A. castellanii* and *A. polyphaga* has been widely studied, and the pathogenic mechanisms remain unknown. Thus, we examined 7 clinically pathogenic strains by comparative genomic, phylogenetic, and rhizome gene mosaicism analyses to explore amoeba–symbiont interactions that possibly contribute to pathogenesis. Genetic characterization and phylogenetic analysis showed differences in functional characteristics between the "open" state of T3 and T4 isolates, which may contribute to the differences in virulence and pathogenicity. Through comparative genomic analysis, we identified potential genes related to virulence, such as metalloprotease, laminin-binding protein, and HSP, that were specific to the genus *Acanthamoeba*. Then, analysis of putative sequence trafficking between *Acanthamoeba* and Pandoraviruses or *Acanthamoeba castellanii* medusaviruses provided the best hits with viral genes; among bacteria, *Pseudomonas* had the most significant numbers. The most parsimonious evolutionary scenarios were between *Acanthamoeba* and endosymbionts; nevertheless, in most cases, the scenarios are more complex. In addition, the differences in exchanged genes were limited to the same family. In brief, this study provided extensive data to suggest the existence of HGT between *Acanthamoeba* and ARMs, explaining the occurrence of diseases and challenging Darwin's concept of eukaryotic evolution.

**IMPORTANCE** *Acanthamoeba* has the ability to cause serious blinding keratitis. Although the prevalence of this phenomenon has increased in recent years, our knowledge of the underlying opportunistic pathogenic mechanism maybe remains incomplete. In this study, we highlighted the importance of *Pseudomonas* in the pathogenesis pathway using comprehensive a whole genomics approach of clinical isolates. The horizontal gene transfer events help to explain how endosymbionts contribute *Acanthamoeba* to act as an opportunistic pathogen. Our study opens up several potential avenues for

Address correspondence to Yongliang Lou, louyongliang2013@163.com, or Meiqin Zheng, zmqlyllh@126.com.

The authors declare no conflict of interest.

future research on the differences in pathogenicity and interactions among clinical strains.

**KEYWORDS** *Acanthamoeba*, endosymbiont, comparative genome analysis, virulence gene, horizontal gene transfer

*A*canthamoeba are among the most ubiquitous protists and are found in natural or artificial habitats, mostly humid habitats, such as soil, drinking water, air, sediments, and engineered environments (1). These species are widely recognized as causing devastating and debilitating human corneal infectious diseases, named *Acanthamoeba* keratitis (AK), occurring mostly in contact lens wearers. More importantly, the number of cases in non-contact lens wearers in Asian countries has grown in recent years (2, 3). Based on previous studies, *Acanthamoeba* species are distributed into 23 different ribogenotypes (T1–T23) (4–6). Although genotypes of T4 are the most prevalent, AK caused by non-T4 genotypes is associated with more severe outcomes (7). Patients with AK suffer from pain with photophobia and tears, ring-like stromal infiltrate, epithelial defects (8), and similar clinical features and are often misdiagnosed with herpetic, bacterial, or mycotic keratitis, leading to delayed treatment of the disease (9). Moreover, the pathogen can transform from trophozoite form to a double-walled cyst state under harsh conditions, such as changes in temperature and pH or a lack of nutrients. And the cyst has the ability to resist environmental pressures and drug effects to survive for more than 20 years. leading to recurrence (10, 11). The pathogenic cascade of AK involves multivariate factors that are divided into direct and indirect factors, beginning with the amoebae phagocytosing the epithelial cornea by producing specific adhesins and toxins and culminating in melting of the corneal stroma (9, 12). Currently, AK can cause blindness if not treated properly and immediately (13). Therefore, early diagnosis and shortening the clinical course of observation are urgently needed. Nevertheless, research on pathogenic pathways in AK and the potential effects of endosymbionts is incomplete.

*Acanthamoeba* species are phagocytic protists that feed on bacteria, fungi, yeasts, and algae $> 0.5$ $\mu$m in size by means of selectively grazing to regulate the environmental microbial population (14, 15). However, some amoeba-resisting microorganisms (ARMs), including bacteria, fungi, and giant viruses, have acquired the capacity to resist phagocytosis to survive intracellularly and multiply (16–19). Some amoeba-resistant bacteria (ARB) have developed strategies to lyse the amoebal host, resist phagocytosis, and survive intracellularly so as to be considered as endosymbionts. The intracellular lifestyle protects ARMs from chlorine and other biocides when amoebae encysted, more importantly, contributing to develop and maintain virulence traits including antibiotic resistance and adapt to life within human macrophages. This contributes to *Acanthamoeba* being a potential vehicle of virulent human pathogens, such as *Legionella pneumophila* and *Mycobacterium* sp., as reported in previous studies (17, 20). To date, three new families of giant viruses have been identified, namely, *Mimiviridae*, *Marseilleviridae*, and *Lavidaviridae*, along with seven other lineages, including Pandoraviruses, pithoviruses, faustoviruses, Mollivirus sibericum, Kaumoebavirus, cedratviruses and Pacmanvirus (21). Moreover, endosymbionts living in sympatry in *Acanthamoeba* have larger genomes than allopatric ones (22), suggesting that the sympatric lifestyle can increase the chance of horizontal gene transfer (HGT) between endosymbionts and enrich the gene pool.

Compared to Darwin's concept of vertical inheritance playing the foremost role in eukaryotic evolution, HGT represents a faster mechanism for acquiring genetic variability and shaping genomes, which is difficult to achieve through vertical evolution (23, 24). In light of the progress in whole-genome sequencing, more attention has been given to the role of lateral transfer in the constitution of the gene repertoire. To date, several studies have provided evidence to support the occurrence of HGT; for example, analysis of bacterial endosymbionts in clinical isolates from AK patients exhibited the existence of *Pseudomonas*, *Mycobacteria*, *Chlamydia* and *Legionella* species in amoeba

**Table 1** Summary of the genomes for *Acanthamoeba* isolates

| Strain | Genome size (Mb) | Sequence contigs (*n*) | Largest contig (*n*) | GC content (%) | N50 | N75 | Predicted proteins (*n*) | Annotated proteins (*n*) |
|---|---|---|---|---|---|---|---|---|
| WBN | 62.29 | 15,622 | 754,261 | 58.84 | 8,340 | 3,634 | 31,492 | 27,971 |
| ZXY | 47.12 | 15,859 | 287,054 | 58.06 | 5,282 | 2,468 | 21,576 | 18,302 |
| LCH | 31.70 | 23,963 | 214,298 | 57.44 | 1,438 | 907 | 12,838 | 11,024 |
| ZWL | 50.13 | 18,337 | 1,416,858 | 57.91 | 70,244 | 1,733 | 29,151 | 27,465 |
| LYL | 39.22 | 23,431 | 265,319 | 59.00 | 1,991 | 1,115 | 17,120 | 14,865 |
| SNN | 63.34 | 27,520 | 1,008,868 | 57.28 | 5,007 | 1,361 | 32,014 | 29,372 |
| YM | 58.91 | 12,888 | 1,717,570 | 57.37 | 8,655 | 3,565 | 30,292 | 26,774 |

hosts, forming potential interactions. Such interactions were reported to have dual clinical significance related to pathogenesis; on the one hand, *Acanthamoeba* could protect bacterial endosymbionts from hostile environmental conditions and enhance invasiveness and virulence. On the other hand, endosymbionts can also influence the pathogenicity, virulence, or susceptibility to antiamoebic drugs of *Acanthamoeba* (25).

Conversely, the interactions are far more complex than known and are not limited to bacterial endosymbionts. To date, a few studies have contributed to the knowledge of the genomic characteristics of *Acanthamoeba*, whereas only *A. castellanii* and *A. polyphaga* have been investigated in depth (26–29). According to recent findings, HGT has mostly been identified by comparing *Acanthamoeba* genomes to those of endosymbionts (30). In this study, we collected 7 clinical strains from AK patients and 13 publicly available genomes, including pathogenic and nonpathogenic genomes, presenting comprehensive functional and genetic analyses of the T3 and T4 clinical isolates, including phylogenetic and pangenome analyses. We identified virulence factors related to pathogenicity through comparative genomic analysis. Our study also provided much evidence to prove the occurrence of HGT between *Acanthamoeba* and ARMs. In summary, our study determined the presence of endosymbionts in the clinical *Acanthamoeba* host and compared their potential differences in the pathogenesis of the disease.

## RESULTS

**General genomic features of AK isolates and phylogenetic reconstruction.** The high-quality data of 7 isolates of AK pathogenic strains were sequenced and assembled. We used FastQC and Trimmomatic to control the raw data with an average quality above 28, and reads of low quality were excluded. The main obtained genomic characteristics and annotation information are presented in Table 1. The estimated sizes of the 7 draft genomes ranged from 31.7 Mb to 63.34 Mb, and 57.99% of the GC content on average showed little difference among the isolates. The maximum number of sequence contigs shared between isolates was 27,520, and the minimum number was 1,288.

Searching against other *Acanthamoeba* sequences available in public databases showed that most of the genomic lengths were similar, with a size of 66 Mb for the draft genome of *A. triangularis* ATCC 50254 (31), 42 Mb for the genome of *A. castellanii* Neff (26), and 49 Mb for the *A. polyphaga* Linc-AP1 genome (Table 2) (27). Additionally, the GC content, predicted proteins, and annotated proteins of each isolate were similar to those of the above-mentioned amoebae, indicating that all AK pathogenic strains may share a genetic relationship with these amoebae.

To verify the phylogenetic relationships of different *Acanthamoeba* strains, phylogenetic reconstruction based on 18S ribosomal genes was performed (Fig. 1). Only 1/7 strain belonged to the T3 genotype, while 6/7 strains were clustered near genotype T4 of *Acanthamoeba* spp., which, to the best of our knowledge, was the most representative of infections in both keratitis and nonkeratitis samples (32). Moreover, strains of the T4 genotype were clustered in three subtypes (T4A, T4D, T4E), which indicated that they may difference in pathogenicity and virulence. It is worth mentioning that the pathological characteristics of AK caused by different genotypes are different but are poorly studied (33).

**TABLE 2** Comparison with main several amoebas' genomic features

| Organism | Genome size (Mb) | GC content (%) | Predicted proteins (n) | Annotated proteins (n) |
|---|---|---|---|---|
| *Acanthamoeba triangularis* ATCC 50254 | 66 | 58.6 | 37,062 | 33,168 |
| *Acanthamoeba castellanii* Neff | 42 | 57.8 | 20,681 | 15,455 |
| *Acanthamoeba polyphaga* Linc-AP1 | 49 | 58.1 | /[a] | / |
| *Acanthamoeba castellanii* ATCC 50370 | 121 | / | 82,310 | / |
| *Acanthamoeba polyphaga* ATCC 30872 | 124 | / | 47,246 | / |
| *Willaertia magna* C2c Maky | 37 | 25 | 18,519 | 13,571 |
| *Naegleria fowleri* ATCC 30863 | 30 | 35 | 17,252 | 16,021 |
| *Naegleria gruberi* NEG-M | 41 | 33 | 15,727 | 9,090 |
| *Naegleria lovaniensis* ATCC 30569 | 31 | 37 | 15,195 | 13,005 |
| *Dictyostelium discoideum* AX4 | 34 | 22 | 13,541 | 8,422 |
| *Entamoeba histolytica* strain HM-1: IMSS | 21 | 24 | 8,201 | 4,076 |

[a]/, data was not available.

**Pangenomic and functional characterization of AK pathogenic strains.** In view of the lack of research on the pangenome characteristics of *Acanthamoeba*, the core and pangenome of each species were determined for the purpose of comparing the general genetic similarities and diversity among different *Acanthamoeba* species. Empirical power-law regression and exponential curve fitting were used for extrapolation of the pan- and core genome curves, respectively, as presented in Fig. 2A and B. The number of pangenomes had risen dramatically, with a considerable number of new genes identified, indicating that the pangenomes were in an "open" state. Conversely, the core genome decayed in an exponential pattern and tended to find a balance, as shown by the results. In fact, the greater the capacity to acquire exogenous DNA, the more likely it is that the organism harbors an "open" pangenome and has a higher HGT range (34, 35).

Furthermore, detailed analysis of the distribution and function of the core and unique genes of the analyzed genomes was completed (Fig. 2C), and the results showed that a total of 77 clusters consisted of core genes, and the genes with associations and uniqueness apparently varied. Fisher's exact test revealed that among the Cluster of Orthologous Group of Proteins (COG) database categories, core genes were found to be significantly abundant in only the cytoskeleton (FDR = 0.027). However, the accessory genes were mostly enriched in cellular processes and signaling and in metabolism, including amino acid transport and metabolism, posttranslational modification, protein turnover, chaperones, nucleotide transport and metabolism, and energy production and conversion. Unique genes (FDR < 0.05) were enriched in cell wall/membrane/envelope biogenesis and in replication, recombination, and repair. Interestingly, we found a considerable number of unique genes in strains YM ($n = 11,064$) and WBN ($n = 8,390$) and the lowest number of unique genes in LCH ($n = 671$). However, the differences revealed that unique genes of YM were mostly enriched in cellular processes and signaling, and those in WBN were enriched in the metabolism category. In addition, the assigning of many genes to the unknown function ($n = 386$ and $n = 409$, respectively) and general function prediction only ($n = 821$ and $n = 660$, respectively) categories could be explained to some extent by the predicted protein numbers of YM ($n = 30,292$), WBN ($n = 31,492$), and LCH ($n = 12,838$).

Based on the similarities and differences in pangenome functional characteristics, we further compared the enrichment of COG functional diversity with all *Acanthamoeba* genomes isolated from AK patients clustered in T3 and T4. The outcome was similar to that for enriched genes from genotype T3 (YM); the genes were associated with cellular processes and signaling: cell motility and defense mechanisms. Additionally, genes from genotype T4 (WBN, LCH, SNN, ZWL, ZXY, LYL) were found to be significantly enriched in amino acid transport and metabolism, inorganic ion transport and metabolism, and nucleotide transport and metabolism. Moreover, numerous genes were enriched in the unknown function category in the strains (Fig. 2D). To summarize, compared to the over-expression of cellular process and signaling genes in T3 and metabolism genes in T4, the

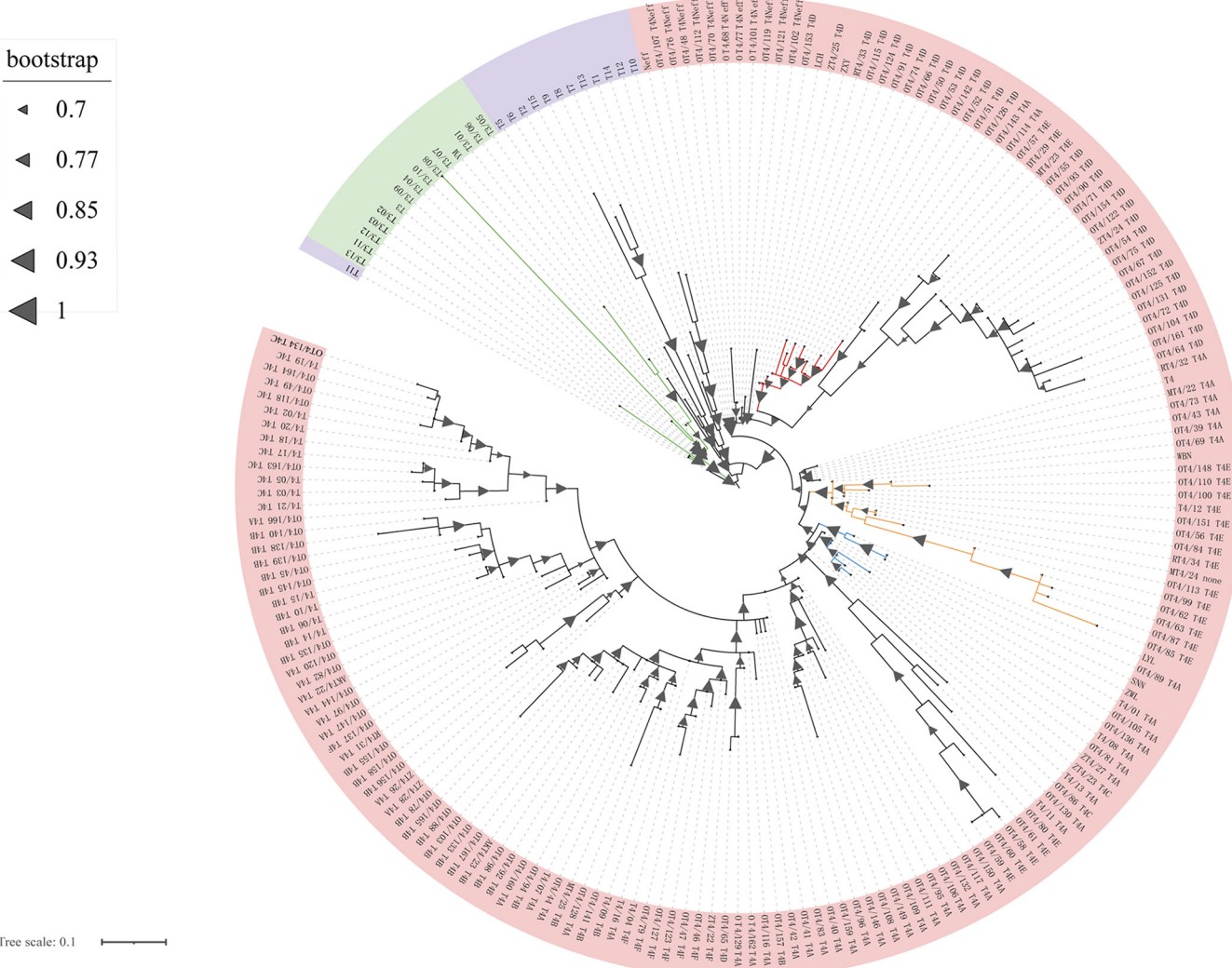

**FIG 1** Phylogenetic analysis based on 18s rDNA among *Acanthamoeba* isolates and diverse genetypes. Different gene types are represented by different colors: red label represents T4 genetype, T3 genetype in green, and other genetypes in purple. Subtype clusters are distinguished by colors. All types are included: T1: *A. castellanii* V006 (U07400); T2: *A. palestinensis* Reich (U07411); T3: *A. griffini* H37 (S81337); T4: *A. castellanii* (U07413); T5: *A. lenticulata* E18-2 (U94735); T6: *A. palestinensis* 2802 (AF019063); T7: *A. astronyxis* R&H (AF019064); T8: *A. tubiashi* OC-15C (AF019065); T9: *A. comandoni* (AF019066); T10: *A. culbertsoni* Lilly A1 (AF019067); T11: *A. hatchetti* BH-2 (AF019068); T12: *A. healyi* (AF019070); T13: *Acanthamoeba* sp.UWC9 (AF132134); T14: *Acanthamoeba* sp.PN15 (AF333607); T15: *A. jacobsi* AC305 (AY262365); *A. castellanii* Neff (U07416).

gene functional diversity could result in HGT among strains, which may influence the spread of virulence and pathogenicity.

**Virulence-related genes in *Acanthamoeba* genomes.** Considering the devastating nature and poor outcomes of AK, a comprehensive study of the molecular pathogenesis associated with the disease is needed. We performed a comparative genomic study with 15 pathogenic *Acanthamoeba* strains and 5 different amoebae that do not cause keratitis to identify the virulence traits of *Acanthamoeba* that may be potential targets for advanced diagnosis and alternative therapeutic interventions. The potential keratitis virulence genes were divided into several categories according to parasite molecules among clusters. Moreover, laminin-binding protein (AhLBP), which mediates the adhesion process, a paramount step in pathogenic cascades (36), was found in only 28.6% (2/7) of pathogenic strains. In terms of proteases, the zinc carboxypeptidase superfamily protein gene (4/7) and other peptidase genes (6/7) have cytotoxic effects on human corneal epithelial cells and keratocytes and support deeper corneal penetration by *Acanthamoeba* (12, 37, 38) in almost all strains of *Acanthamoeba*, except strain ZWL. It is worth mentioning that the genes associated with "lipase," "cytoskeleton," and "glycosidase" were relatively weak

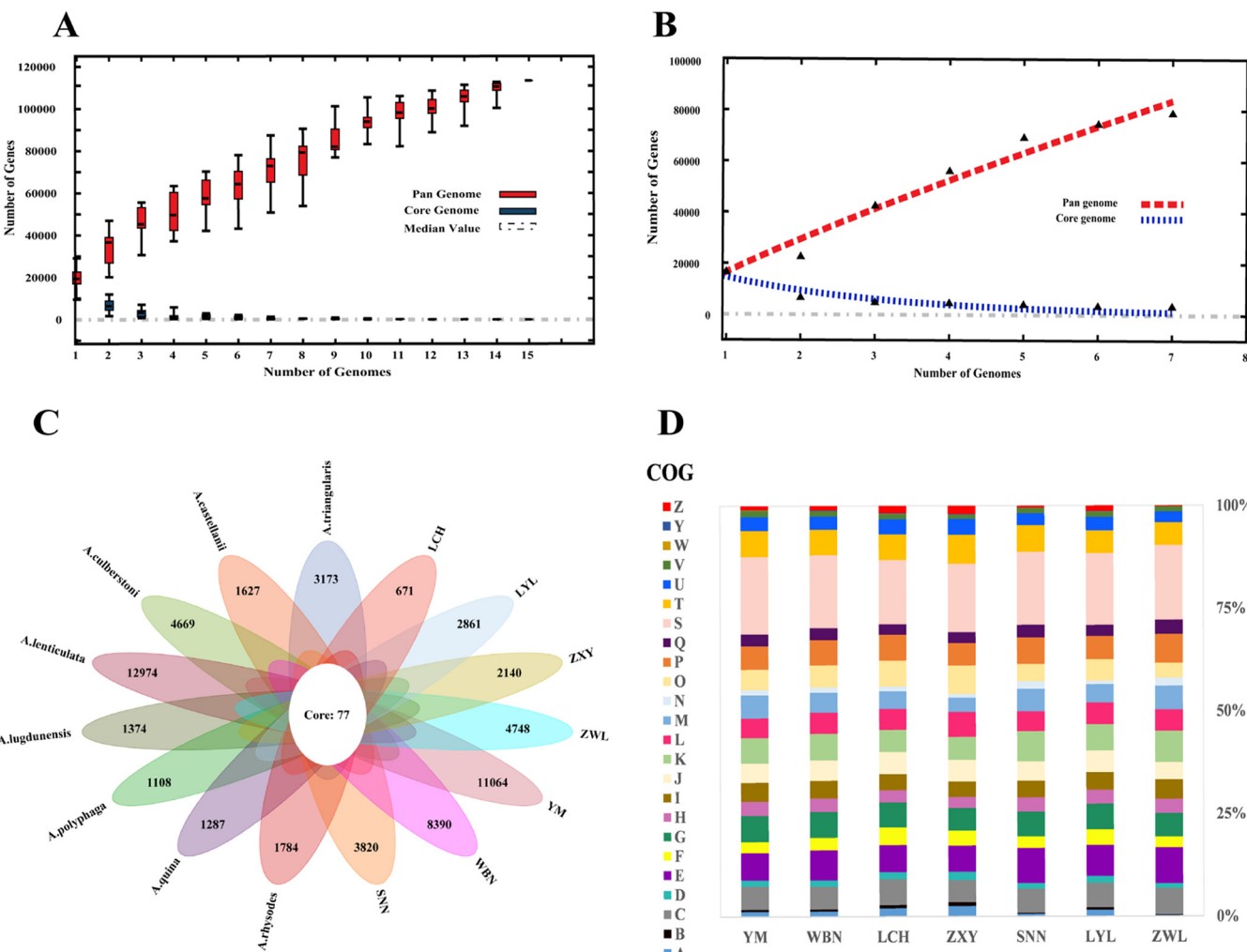

**FIG 2** Pan-genome analysis and COG functional annotations of *Acanthamoeba* species. (A) The pan genome profile trends for pan (red) and core (blue) of collected *Acanthamoeba* species. The accumulation plots display the relationship between core and pan genomes. (B) The pan genome profile trends for pan (red) and core (blue) genomes of isolates. (C) Comparative analysis of *Acanthamoeba* core and pan genomes. (D) COG function annotations for 7 isolates. Involved COG categories are as follows: [A] RNA processing and modification; [B] Chromatin structure and dynamics; [C] Energy production and conversion; [D] Cell cycle control, cell division, chromosome partitioning; [E] Amino acid transport and metabolism; [F] Nucleotide transport and metabolism; [G] Carbohydrate transport and metabolism; [H] Coenzyme transport and metabolism; [I] Lipid transport and metabolism; [J] Translation, ribosomal structure, and biogenesis; [K] Transcription; [L] Replication, recombination, and repair; [M] Cell wall/membrane/envelope biogenesis; [N] Cell motility; [O] Posttranslational modification, protein turnover, chaperones; [P] Inorganic ion transport and metabolism; [Q] Secondary metabolites biosynthesis, transport, and catabolism; [T] Signal transduction mechanisms; [U] Intracellular trafficking, secretion, and vesicular transport; [V] Defense mechanisms; [W] Extracellular structures; [R] General function prediction only; [S] Function unknown; [Y] Nuclear structure; [Z] Cytoskeleton.

conserved among strains. Interestingly, the results showed that the number of virulence-related genes within different draft genome sequences of *Acanthamoeba* strains was almost consistent with the phylogenomic analyses; for instance, the strains ZWL, SNN, and LYL were significantly less abundant than the other strains (Table S1 in the supplemental material). Then, we chose a representative isolate strain of WBN to illustrate mainly the virulence genes involved in host invasion (Table 3).

For simplicity, the primary pathogenic factors causing keratitis were divided into direct and indirect classes (9). First, we detected the presence of AhLBP, a protein that participates in the adherence of *Acanthamoeba* to corneal epithelial cells, particularly in the intercellular space. This binding indicated that the process continued with secondary processes such as cytolysis, phagocytosis, and induction of apoptosis (37). The survey showed that the presence of 4 genes related to the cytoskeleton, especially 1 gene encoding myosin, was confirmed to play an essential role in the pathogenesis of AK by actin-mediated cytoskeletal rearrangement (39, 40). Correspondingly, 2 genes

**TABLE 3** Potential virulent factors from representative strain involved in host invasion[a]

| Parasite molecules | Gene identification | Function |
|---|---|---|
| Adhesion | g20836 | AhLBP |
| Cytoskeleton | g8419 | Myosin |
| Phagocytosis | g22433 | Protein tyrosine kinase |
| Lipase | g16565 | Phospholipase D |
| | g16127 | Type-B carboxylesterase lipase family |
| Metalloprotease | g17909 | Zinc carboxypeptidase superfamily protein |
| CPs | g25661 | Papain family cysteine protease subfamily protein |
| | g24707 | Cysteine protease 3 |
| Peptidase | g8731 | Microsomal signal peptidase 25 kda subunit |
| | g24644 | Peptidase, S8/S53 subfamily protein |
| | g26480 | Peptidase C19 family |
| Glycosidase | g16880 | Glycosyl hydrolases family 15 |
| Antioxidant defense | g15744 | Oxidoreductase |
| Ecto-ATPases | g27780 | ATPase family associated with various cellular activities (AAA) |
| Superoxide dismutase | g9157 | Superoxide-generating NADPH oxidase activator activity |
| Temp tolerance | g16215 | HSP20-like chaperone |
| | g12895 | Heat shock 70 kDa protein |

[a]AhLBP: laminin-binding protein; CPs: cysteine protease.

related to the cyclin family were also identified. Following adhesion and breakdown of the corneal epithelium, the process of stromal invasion is mediated by secretion of metalloproteinase and serine and cysteine proteinases, as reported. We identified 1 protein tyrosine kinase related to phagocytosis, 2 lipases, including 1 phospholipase, 1 metalloprotease belonging to the zinc carboxypeptidase superfamily, 3 other genes encoding peptidases involved in host invasion, and 2 cysteine proteases (CPs). In addition, we identified the presence of glycosidase and Ecto-ATPase, which generated the resultant ADP, exerting toxic effects on host cells in a contact-independent mechanism. The antioxidant enzymes oxidoreductase and superoxide dismutase are also involved in amoebal defense against reactive oxygen species. Among indirect factors, we identified 2 genes encoding heat shock proteins (HSP20 and HSP70) that enhanced the ability of the cells to grow at high temperatures and were potential indicators of pathogenicity (2, 41).

**Taxonomic distribution among *Acanthamoeba* genes.** To predict the likelihood of the occurrence of HGT for gene trafficking between the amoebae and ARMs, we analyzed the taxonomic distribution of the proteins of AK pathogenic strains, which were assessed based on the best BLAST hits. The results indicated that the ARMs present within strains, including giant viruses, bacteria, and fungi, exhibited larger genomes than their mammal-infecting relatives (42). The proteins belonging to *A. castellanii* str. Neff accounted for a large proportion of the draft genome. Based on the large number of ARMs best matched with *Acanthamoeba*, we could infer the existence of important gene trafficking between *Acanthamoeba* and the infecting ARMs.

First, endosymbiont genes from *Klebsiella*, *Burkholderia*, *Acinetobacter*, *Bacteroidetes* bacterium, and *Chlamydiae* were identified in all the strains (Fig. 3). Then, we found that *Pseudomonas* spp. (6/7) provided the greatest numbers of best hits with *Acanthamoeba* among ARB; these species are commonly responsible for acute-onset and highly destructive keratitis (43), and the result was consistent with previous findings from clinical isolates (25). Furthermore, *Pseudomonas aeruginosa* genes were found in 71.4% (5/7) of pathogenic strains. Analysis of the presence and conservation of these genes in the draft genome sequences showed that some existed in a majority of genomes, while some were present in only a few genomes. For instance, 6 of 7 isolates with endosymbionts had sequences similar to those of bacteria in the *Mycobacterium* genus, whereas the proteins shared with the best hits belonged to *Mycobacterium tuberculosis* in 42.9% (3/7) of strains. Endosymbiont sequences belonging to *Rickettsia* were also detected in 3/7 isolates. In

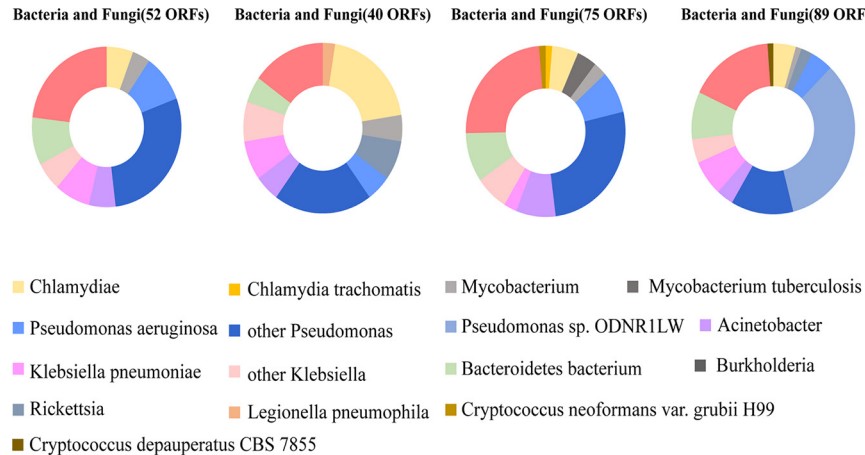

**FIG 3** Taxonomic distribution of the predicted bacteria and fungi proteins in representative strains. The strains successively consisted of three genotype T4 (WBN, ZXY, ZWL) and one genotype T3 (YM) classified in different branches regarding phylogenetic. The number of predicted proteins was indicated.

addition, we found only 1 gene that was best matched with *Legionella pneumophila* (strain ZXY), a pathogenic human bacterium that causes respiratory illness, and *Chlamydia trachomatis* (strain ZWL), which is responsible for the occurrence of sight-threatening trachoma. Finally, 3 genes of different isolates (YM, LYL, ZWL) were observed to be segregated into different clusters: *Cryptococcus depauperatus* CBS 7855, *Cryptococcus neoformans* var. *grubii* H99. and *Cryptococcus neoformans* var. *grubii* Bt1 (Table S2).

With regard to amoeba-resistant viruses (ARVs), the results appeared to be similar (Fig. 4). Compared with the 7 strain genomes that we analyzed, we found that most of the viral sequences shared with those of Pandoraviruses (7/7) and *Acanthamoeba castellanii* medusavirus (6/7), which were isolated from hot spring water and survived on *Acanthamoeba castellanii* (44). In fact, the Pandoravirus members that we detected were clustered in 8 Pandoravirus strains (*P. quercus*, *P. inopinatum*, *P. macleodensis*, *P. celtis*, *P. neocaledonia*, *P. salinus*, *P. dulcis*, and *P. japonicus*). Genes from Mollivirus sibericum, Mollivirus kamchatka, Marseillevirus and Pithovirus sibericum were found in 42.9% (3/7), 57.1% (4/7), 28.6% (2/7), and 28.6% (2/7) of isolates, respectively. Four genes shared homologous sequences with members of *Mimiviridae*, including Moumouvirus monve, Moumouvirus australiensis, Pacmanvirus A23, and Catovirus CTV1, which were present in different strains (strains YM, ZXY, ZWL). In addition, 4 genes from *Phycodnaviridae* were present alone, which is rarely observed (Table S3).

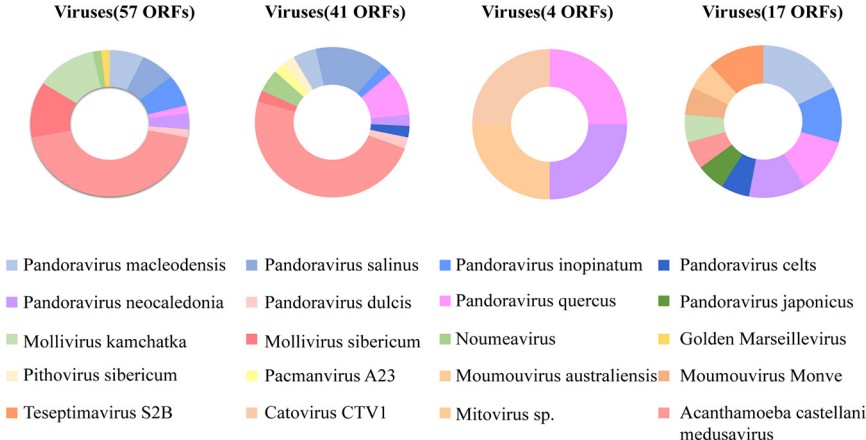

**FIG 4** Taxonomic distribution of the predicted viral proteins in representative strains. The strains successively consisted of three genotype T4 (WBN, ZXY, ZWL) and one genotype T3 (YM) classified in different branches regarding phylogenetic. The number of predicted proteins was indicated.

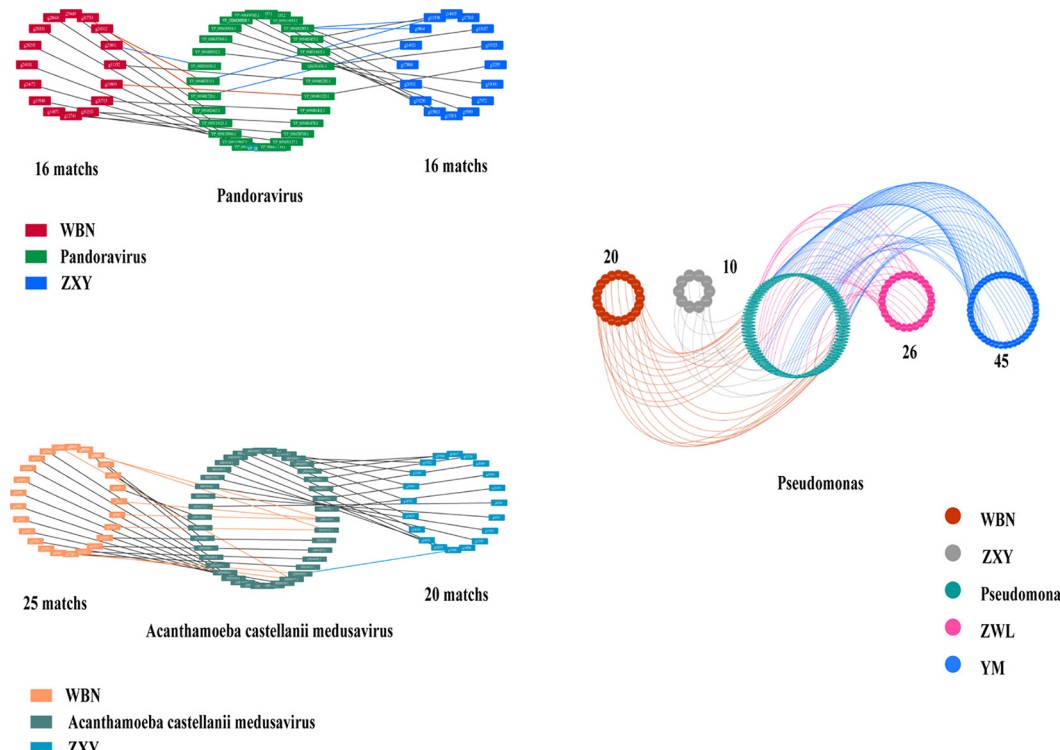

**FIG 5** Network of isolates with representative homologous genes. (Left) Giant viruses genes of Pandoraviruses and *Acanthamoeba castellanii medusavirus*. (Right) Bacteria genes of *Pseudomonas*, and the number of exchanged genes for which a homolog was identified in each isolate, represented by different colors.

The numbers and nature of the potentially transferred genes were highly variable among different families; however, a majority of these genes remained conserved within a given family. To investigate the diversity among strains, we compared the sequences detected in strains WBN and ZXY with those from representative homologous giant viruses (Pandoraviruses and *Acanthamoeba castellanii* medusavirus) and observed little difference within the same family (Fig. 5). In summary, there is little overall difference in HGT within families among strains, but the considerable diversity between the different strains may lead to differences in potential interactions with ARMs. The genes absent and present in various species of *Acanthamoeba* may be related to phenotypic differences and pathogenic diversity.

**Possible horizontal gene transfer between *Acanthamoeba* and endosymbionts.** Based on the above analysis, the need to prove the existence of important gene trafficking between *Acanthamoeba* and infecting ARMs is urgent. Thus, we reconstructed the phylogeny to assess possible HGT between giant viruses and *Acanthamoeba*. For all patterns of HGT, we confirmed that, in at least three cases, genes were transferred from giant viruses to *Acanthamoeba*, which was proven to be the most concise evolutionary scenario regardless of other existing situations (Fig. 6A to C). Moreover, we also found at least three cases in which transfer occurred in the opposite direction, from *Acanthamoeba* to giant viruses (Fig. 6D–F).

While the drawbacks of phylogenetic analyses and nucleotide sequence transfer contributed to the inadequate results, we conducted an in-depth analysis of the predicted sequence transfer in the above six cases, comprising two transfers in opposite directions by means of mosaicism. The most similar homologs shared with each sequence of all *Acanthamoeba* isolates were identified in a more comprehensive manner. Finally, we observed sequence mosaics in all cases; that is, all the best homologous sequences came from different origins, including fungi, bacteria, archaea, and viruses (Fig. 7). Based on the complexity of the large number of homologous sequences and

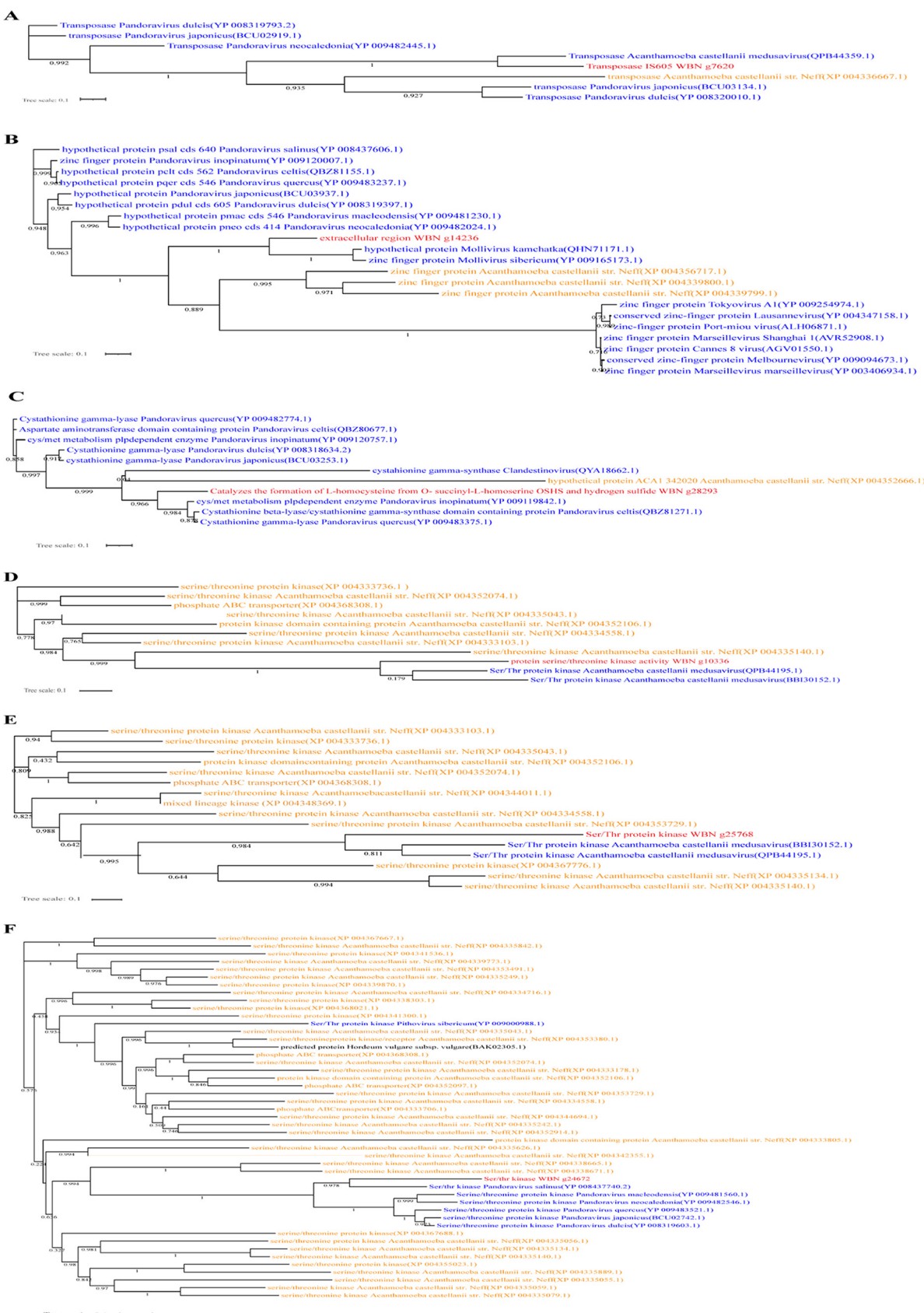

**FIG 6** Representation of phylogenetic analysis for six cases in *Acanthamoeba* isolates with giant viruses origin homologous. The tree was performed based on homologous sequences acquired from searching against the nr database by BLASTp. The horizontal gene transfers

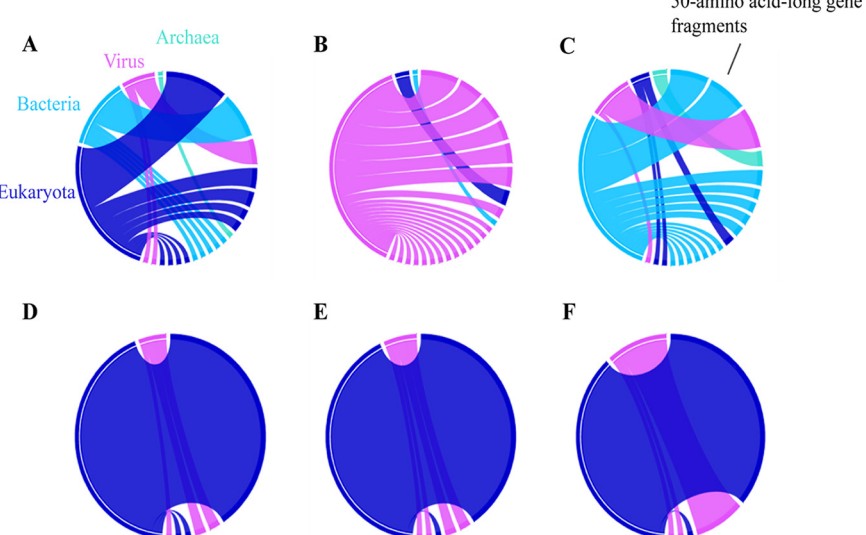

**FIG 7** Rhizomes gene mosaicism of *Acanthamoeba* sequences shared homologs with giant viruses. The six cases of A–F correspond to the above six cases. We searched for the 50 best homologous sequences for each *Acanthamoeba* gene sequence, and classified them into bacteria, viruses, fungi, and archaea according to their origin, and integrated them in a circular visualization.

sequence mosaics between *Acanthamoeba* and virus sequences, it can be inferred that they also interact with other nonoptimal homologous virus organisms, which can improve the interpretation of the phylogenetic analysis results.

## DISCUSSION

*Acanthamoeba* is the most common causative agent of AK, which is a painful and progressive ocular disease associated with trauma (2). It not only acts as a predator that feeds on the microbial population by phagocytosis to control the microbial communities, but also as a reservoir and vehicle of susceptible hosts (15, 45). Currently, a comprehensive understanding of the genomes of pathogenic *Acanthamoeba* strains is lacking. In addition, research on clinical isolates is also deficient. Thus, this is the most comprehensive analysis of the genomes of pathogenic *Acanthamoeba* strains isolated from patients with AK and publicly available genomes of amoebae through several approaches, including comparative genomic, phylogenetic, and sequence network analyses, with the aim of revealing genetic similarities and differences among pathogenic and nonpathogenic strains as well as different gene types.

In our study, we compared the genomic characteristics of pathogenic isolates with those of other amoebae from public databases. We explored the genomic content (in terms of genome size, sequence contigs, predicted proteins, and GC content) of pathogenic isolates that shared high similarity with the pathogenic *Acanthamoeba* genome (*A. triangularis* ATCC 50254, *A. castellanii* Neff, and *A. polyphaga* Linc-AP1). It is surprising that the sizes of the *A. castellanii* ATCC 50370 (121 Mb) and *A. polyphaga* ATCC 30872 (124 Mb) genomes assembled in 2018, which improved the potential overestimation and filtered very short sequences assembled in 2015 (accession: PRJEB7687), were also approximately at least 2-fold larger than those that we obtained. These differences in sizes may be explained by differences in the sequencing technology and assembly tools used, and we hope that this issue can be clarified in future studies. The "open" state of genomes, especially for species living in bacterial communities, such as coagulase-negative staphylo-

**FIG 6** Legend (Continued)

were from giant viruses to *Acanthamoeba* (A–C), and in the reverse way from *Acanthamoeba* to giant viruses (D–F). In red: protein of *Acanthamoeba* isolates; in blue: homologs from viral family; in orange: homologs from other *Acanthamoeba* species; in black: homologs from other organisms.

cocci (46), indicated that the phenomenon of potential HGT within endosymbionts exists in *Acanthamoeba* species. Consistent with previous studies, in *Acanthamoeba* species that cause human infections, the most common causative genotype is T4, followed by genotype T3, based on the phylogenetic tree (47). Furthermore, the genomes belonging to different types of T4 also have a difference of 0–4% in sequences and 6%–12% in gene types (48). In our study, we found that 3 isolates belonged to subtype T4A, which was the most frequent subtype of genotype T4 and confirmed having subtype diversity of alleles within the DF3 region of the gene. It is reported that alleles would become differentiated during evolution by mutations that independently occurred within the separate lineages of each subtype (49). Thus, each subtype may have ancestral genes limited to one subtype that contribute to pathogenic diversities. And we reported a rare comparative analysis of gene sequence differences between T3 and T4 because strains with various virulence traits also contribute to failure in the development of effective chemotherapeutic agents for AK (7).

Searches for predicted proteins against the NCBI GenBank protein sequence (nr) and COG public databases allowed us to perform detailed biological function annotation and identify diverse putative origins of pathogenic *Acanthamoeba* strains. A large number of genes predicted from the genome sequences had no homologs in the nr database, suggesting that numerous genes have not been mined. Through comparative genomics, we found evidence of potential pathogenic genes linked to keratitis. It is generally believed that the adhesion of *Acanthamoeba* to the cornea is a crucial prerequisite for the subsequent inflammatory response, and the degree of adhesion is directly proportional to the strength of the host's inflammatory response (50). AhLBP participates in the initial phase, and infiltration is limited to the corneal epithelium. Therefore, the selectivity of *Acanthamoeba* for the host cornea also determines the differences in the specificity of AK in different hosts. Moreover, it is worthy to note that mannose binding protein, which is considered another critical gene for corneal adhesion, is not found based on our analyses. This issue deserves clarification in future studies. N-terminal-domain-containing proteins belonging to the cyclin family were identified in our research and may explain how the adhesion of *Acanthamoeba* to host cells regulates the expression of a number of genes important for the cell cycle, such as cyclins F and G1 (51). The myosin light chain, which is partially inhibited by a Rho kinase inhibitor (Y27632) to block stress fiber formation, indicates the importance of actin-mediated cytoskeletal rearrangement in the phagocytosis pathway. Furthermore, pathogenic *Acanthamoeba* species exhibit increased extracellular protease activity. These proteases produce a potent cytopathic effect to kill host cells and degrade the epithelial basement membrane as well as the stromal matrix to progress into deeper layers of the cornea (37). In our findings, we not only identified various peptidases belonging to different families but also identified cysteine proteases and metalloproteases of unknown origin. Recently, with the preliminary elucidation of mechanisms of actions of proteases at the molecular level, the potential applications of proteases as therapeutic targets have increased, as evidenced by the use of protease inhibitors to treat hypertension and AIDS. The role of phospholipase in membrane disruption, producing host cell damage or inducing inflammatory responses, facilitates *Acanthamoeba* virulence (9). Thus, comparative genomic analysis revealed the potential pathogenicity-related genes, which will help in the development of a new therapeutic approach.

It is well established that amoebae serve as fertile ground for genetic exchange among endosymbionts, which is called the "melting pot" hypothesis (52). Compared with isolated populations, microorganisms living sympatrically in large communities are more prone to exchange sequences between phylogenetically disparate organisms residing within the same amoebal host cell and with the host (17, 22). This means that amoebae serve as both an "intracellular arena" of sequence exchange for microorganisms living within them and a participant in these games. According to previous studies, many of these exchanged genes enriched the repertoire of amoebozoan genomes in a number of important areas, including transport systems, antibiotic resistance, stress responses, bacterial virulence and signaling, pattern recognition, and accumulation of a

substantial gene armory for the purpose of competitively surviving with other amoebae and influencing pathogenicity (26, 53, 54). In this work, we identified several endosymbionts, such as *Chlamydiae*, *Mycobacterium*, *Pseudomonas*, *Legionella*, *Burkholderia*, and *Rickettsia*, and *Pseudomonas* species, that were significantly different among the isolated strains. Through further comparative analysis, we found that *Pseudomonas* species were the most common endosymbionts in all the strains. Combining the analogous analysis results from clinical isolates indicated the same phenomenon, that is, the high abundance of *Pseudomonas* proved that corneal pathogenic bacteria are more pathogenic than *Legionella* (25, 43). Therefore, there is more reason to suspect that the abundance of *Pseudomonas* in pathogenic strains is closely related to virulence and pathogenicity. Of course, more data on virulence factors, clinical outcomes, and drug susceptibility and experiments are needed to verify this in the future.

Among the numerous completely unexplored endosymbionts, we identified members of the *Phycodnaviridae* and *Mycobacterium* families, which were reported as endosymbionts in environmental *Acanthamoeba* species in a previous report (55) and have rarely been reported in clinical samples. It is surprising that *Burkholderia* species are quite common in free-living amoebae compared with *Chlamydiae*, which may reflect a lower affinity of *Burkholderia* endosymbionts for clinically relevant amoebae. Furthermore, we provided considerable evidence of gene exchange among human pathogens in which the occurrence and development of the disease is possibly related to LGT, including *Legionella pneumophila*, *Chlamydia trachomatis*, and *Cryptococcus*. Although this analysis is based on a minor clinical sample size, it provides a practical reference for us to explore the mechanism of AK, and more clinical sample analysis will be carried out in the future to consummate our results.

In the majority of cases, the significance of the horizontal transfer of the sequences cannot be determined due to an insufficient number of matches. At the same time, the obsolescence of Darwin's concept of vertical inheritance and the emergence of the importance of LGT have led to the proposal of the "rhizome of life" as a representative of species evolution and mosaicism of bacterial genomes (56). This global analysis of the whole genomes from eukaryotes, bacteria, archaea, and giant viruses as best matches provides more comprehensive information regarding gene trafficking between amoebae and endosymbionts than phylogenetic trees. In our study, the transfer between amoebae and endosymbionts was the simplest case, and the existence of other conditions cannot be ruled out. Further rhizome analysis showed that sequence exchange is not a one-way process but a complex multidirectional mechanism, that is, interaction with organisms other than the host can occur due to the sympatric lifestyle. Moreover, the number and nature of exchanged genes are still limited to the same family and vary among different families. Comparison with the phylogenetic analysis results showed our results were consistent with the previous hypothesis that a decrease in phylogenetic distance corresponds to an increase in the level of genome conservation, which was also confirmed in the virulence analysis (30).

**Conclusions.** In this study, we performed a comprehensive whole-genome analysis of clinical pathogenic AK strains. Genes of the T3 strain were significantly enriched in cellular processes and signaling, while those of T4 were enriched in metabolic functions, which may influence the differences in virulence in AK. This work provides improved knowledge on the interactions between *Acanthamoeba* and their endosymbionts, highlighting the fact that gene flow is not only a one-way mechanism but a complex multidirectional process in most cases. In particular, *Pseudomonas* species are suspected to hold great significance for pathogenicity among strains. Overall, our study opens up several potential avenues for future research on the differences in pathogenicity and interactions among clinical strains, explaining phenotypic differences and revealing new targets for treatment and prevention of this disease.

## MATERIALS AND METHODS

**Strains.** Seven clinical *Acanthamoeba* isolates from corneal AK ulcer patients were provided by the Department of Laboratory Research, Eye Hospital of Shandong First Medical University, Shandong, China.

Thirteen amoeba genome sequences were publicly available on the NCBI website (http://www.ncbi.nlm.nih.gov) for comparative genomic analysis, including the *Acanthamoeba* species that are the foremost risk factors for AK (CDFF01000001.1: *Acanthamoeba culberstoni*; NAVB01000001.1: *Acanthamoeba lenticulata*; LQHA01000001.1: *Acanthamoeba polyphaga*; CDFB01000001.1: *Acanthamoeba lugdunensis*; CDFN01000001.1: *Acanthamoeba quina*; CDFC01000001.1: *Acanthamoeba rhysodes*; CDFL01000001.1: *Acanthamoeba castellanii*; CACVKS010000000: *Acanthamoeba triangularis*), a pathogenic amoeba that does not cause keratitis (GCA_000499105.1: *Naegleria fowleri*) and nonpathogenic amoebae (PRJEB30797: *Willaertia magna*; GCA_0004985.1: *Naegleria gruberi*; GCA_003324165.1: *N. lovaniensis*; PRJNA13925: *Dictyostelium discoideum*).

**Culture, DNA isolation, and genotyping.** Cultivation of the isolates was performed at 30°C on nonnutrient agar (NNA) plates layered with *Escherichia coli* (ATCC25922) and containing Page's modified Neff's amoeba saline (PAS: 1.2 g of NaCl, 0.04 g of MgSO4 × 7H$_2$O, 0.03 g of CaCl2, 1.42 g of NaHPO4, and 1.36 g of KH2PO4 in 1 L ddH2O) (57). The plates were examined daily, and the trophozoites were harvested in the exponential growth stage. DNA was extracted using the Qiamp DNA blood and tissue kit (Qiagen) (58).

Amplification and sequencing of 18S rDNA with the primers JDP1 and JDP2 (JDP1: 5-GGCCCAG ATCGTTTACCGTGAA-3'; JDP2: 5-TCTCACAAGCTGCTAGGGAGTCA-3') were performed to authenticate *Acanthamoeba* (59). Based on the sequencing outcomes, we performed a phylogenetic analysis of these nucleotide sequences and available gene sequences in the database. All allelic sequences characterizing T4 and T3 types were downloaded from the website (60). Nucleotide sequence alignments were performed by MAFFT (61), and a phylogenetic tree was constructed with FastTree software (62). The phylogenetic tree was visualized and embellished using iTOL v6 online.

**Sequencing, genome assembly, gene prediction, and functional annotations.** Seven DNA samples were prepared for whole-genome sequencing by Berry Genomics Co., Ltd., Beijing, China, using Illumina Technology. First, FastQC (v0.11.9) was used to evaluate the quality of the raw data, and Trimmomatic-0.38 (63) was used to trim the genome sequences by removing low-quality sequences. Then, all the DNA reads were assembled by SPAdes (v3.14.1), and quality assessment of the genome sequences was performed using QUAST (v4.6.0).

Gene prediction in the 7 assembled genome sequences and 13 amoeba genomes from public databases was performed using AUGUSTUS (v3.4.3) software optimized for eukaryotes (64). For functional annotation, the predicted proteins were analyzed with public databases, including the NCBI GenBank protein sequence database (nr) and Cluster of Orthologous Group of Proteins (COG) database. Briefly, to identify homologous sequences and biological functions, the BLASTp (2.10.0+) program was performed against the nr database with an E-value threshold of 1e-03 and diamond parameters (65). COG functional enrichment of the predicted proteins was performed in the EggNOG database via eggnog-mapper (v2.0).

**Pangenome analysis.** Pangenomic analysis of each *Acanthamoeba* species that emerged from the two types, including the 7 isolates and 8 public database sequences, was identified by BPGA (v1.3) by running USEARCH for the fastest clustering (using default parameters, 50% sequence identity cutoff) (66). The predicted protein sequences were the input files obtained from AUGUSTUS. Further analysis of the gene accumulation curve and core-unique sequence composition was based on these findings.

**Analysis of virulence related genes.** To reveal the virulence genes that were probably relevant to AK, amoeba species that were not involved in AK were included to perform comparative genomic analysis. All *Acanthamoeba* spp. that we selected were proven to be pathogenic in AK. First, gene prediction was performed for all amoebae, including the 7 isolates, 8 *Acanthamoeba* spp., 4 nonpathogenic amoebae, and 1 pathogenic amoeba. Then, Proteinortho (v6.0.30), a method to identify orthologous genes, was employed to analyze the 7 isolates and 13 other amoeba species using an e-value ≤ 1e–4 as the threshold, and only genes with a coverage higher than 60% and an identity higher than 50% were considered significant (67). To further investigate the pathogenic mechanism of AK, virulence genes were defined as homologous genes that were present in 9 *Acanthamoeba* species (*A. culberstoni*, *A. lenticulata*, *A. polyphaga*, *A. lugdunensis*, *A. quina*, *A. rhysodes*, *A. castellanii*, *A. triangularis*, and one analyzed isolate) but not in the other amoebae. The functions and distributions of the genes were compared among all the isolates.

**Taxonomical distribution and horizontal gene transfer.** According to BLASTp functional annotation and sequence homology, the taxonomic distribution was determined based on predicted proteins matched with ARMs in the NCBI nonredundant (nr) protein sequence database. Additionally, networks between protein sequences from two giant viruses (Pandoravirus and *Acanthamoeba castellanii* medusavirus) as well as *Pseudomonas* and the genome sequences of clinical isolates were produced with Cytoscape (v3.8.2) (68). For proteins that were proven to have significant hits, the giant virus, bacterial, or fungal sequences were subjected to phylogenetic analysis to confirm the highest level of sequence similarity with an ARM homolog. Protein sequences with insufficient numbers of hits were excluded. The protein sequences were aligned by MUSCLE, and phylogenetic trees were generated using FastTree. Finally, genes belonging to the isolated strains and those with the best hits in the giant viruses were subjected to interactive mosaic graph analysis. This information was obtained from BLASTp searches against the nr protein sequence database of these genes with a window of 50 amino acids. Visualization of the mosaic graph was carried out by Circos.

**Statistical analyses.** COG functional enrichment in core, accessory, and unique genes of all isolates and differences in the functional characteristics of genotypes T3 and T4 were compared using Fisher's exact test and FDR's correction of *P* values, with $P < 0.05$ considered significant. All statistical analyses were carried out by the R package (version 4.0.5).

**Data availability.** The whole genome sequencing data have been submitted to the NCBI Sequence Read Archive (SRA) under BioProject accession number PRJNA817853.

## SUPPLEMENTAL MATERIAL

Supplemental material is available online only.

**SUPPLEMENTAL FILE 1,** XLSX file, 0.02 MB.
**SUPPLEMENTAL FILE 2,** XLSX file, 0.03 MB.
**SUPPLEMENTAL FILE 3,** XLSX file, 0.02 MB.

## ACKNOWLEDGMENTS

We declare that we have no competing interests.

Xiaobin Gu: designed and performed the experiments, data analysis, visualization, and writing (manuscript); Xiuhai Lu: conceived and designed the experiments; Shudan Lin and Xinrui Shi: data analysis and visualization; Yue Shen, Qingsong Lu, and Yiying Yang: performed the experiments; Jing Yang, Jiabei Cai, and Chunyan Fu: data analysis; Yongliang Lou: conceptualization, data curation, resources, and writing (review and editing); Meiqin Zheng: conceptualization, supervision, data curation, funding acquisition, and writing (review and editing).

All authors have seen and approved the final version of the manuscript being submitted.

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
