## [Reviewer comments · Microbiology Spectrum]

Microbiology Spectrum

A comparative genomic approach to determine the virulence factors and horizontal gene transfer events of clinical *Acanthamoeba* isolates.

Xiaobin Gu, Xiuhai Lu, Shudan Lin, Xinrui Shi, Yue Shen, Qingsong Lu, Yiyang Yang, Jing Yang, Jiabei Cai, Chunyan Fu, Yongliang Lou, and Meiqin Zheng

Corresponding Author(s): Meiqin Zheng, Wenzhou Medical University

Review Timeline:

Submission Date:	January 6, 2022
Editorial Decision:	January 26, 2022
Revision Received:	March 22, 2022
Accepted:	March 22, 2022

Editor: Tim Downing

Reviewer(s): Disclosure of reviewer identity is with reference to reviewer comments included in decision letter(s). The following individuals involved in review of your submission have agreed to reveal their identity: Kirti Megha (Reviewer #1)

Transaction Report:

DOI: <https://doi.org/10.1128/spectrum.00025-22>

January 26, 2022

Dr. Meiqin Zheng
Wenzhou Medical University
Chashan University Town
Wenzhou
China

Re: Spectrum00025-22 (A comparative genomic approach to determine the virulence factors and horizontal gene transfer events of clinical *Acanthamoeba* isolates.)

Dear Dr. Meiqin Zheng:

Thank you for submitting your manuscript to *Microbiology Spectrum*. When submitting the revised version of your paper, please provide (1) point-by-point responses to the issues raised by the reviewers as file type "Response to Reviewers," not in your cover letter, and (2) a PDF file that indicates the changes from the original submission (by highlighting or underlining the changes) as file type "Marked Up Manuscript - For Review Only". Please use this link to submit your revised manuscript - we strongly recommend that you submit your paper within the next 60 days or reach out to me. Detailed instructions on submitting your revised paper are below.

Link Not Available

Thank you for the privilege of reviewing your work. Below you will find instructions from the *Microbiology Spectrum* editorial office and comments generated during the review.

Sincerely,

Tim Downing

Editor, *Microbiology Spectrum*

Journals Department
Reviewer comments:

Reviewer #1 (Comments for the Author):

The research article by Gu et al. examined seven clinically proven pathogenic strains of *acanthamoeba* and compared them with those of *acanthamoeba* and other amoebae from public databases to study genomic, phylogenetic, and identification of virulence genes specific to *acanthamoeba*. The authors also demonstrated rhizome gene mosaicism analyses to explore amoeba symbiont interactions that possibly contribute to pathogenesis. Though the manuscript is interesting, well-written, and easy to read and understand, I still have some suggestions/ concerns that need to be addressed before this manuscript is published. My suggestions/concerns are listed below:

1. Line 65-66: To date, 23 genotypes of *acanthamoeba* are known, i.e., T1-T23. According to a recent study, *acanthamoeba* has been isolated from public freshwater sources in Thailand and named as *A. bangkokensis*, and it belongs to the T23 genotype. (doi: 10.1038/s41598-021-96690-0) that need to be updated in the introduction section.
2. Line 82-83 what does mean "selective grazing growth." As per my understanding, it reflects phagocytosis. So my question is

phagocytosis in acanthamoeba specific or non-specific?

3. Line 85-86: How can acanthamoeba resist phagocytosis? It is reported that endosymbiont survives intracellularly inside acanthamoeba and resists acanthamoebacidal digestion. However, I can't understand how acanthamoeba can resist phagocytosis.

4. Italic acanthamoeba and all bacteria names throughout the text.

5. Line 121 authors mentioned that they had studied the "prevalence of endosymbionts." However, results are nowhere mentioned and discussed, and seven is a minimal sample size to determine prevalence.

6. In the material and method, it is mentioned that to culture acanthamoeba, non-nutrient agar method was used, i.e., monoxenic culture, which includes cultivation of acanthamoeba in the presence of bacteria such as E.coli. How about axenic culture .i.e. without any bacteria? For all genomic studies, DNA extraction should be done with pure culture. DNA extracted from monoxenic culture probably has E.coli DNA as well. Justify it

7. Line 353: write the full form of ARB and ARVs

8. A horizontal gene transfer study was done with new and publicly available data. Have authors tried to validate the same with in vitro experiments

9. Line 475-477 authors have identified several endosymbionts, such as Chlamydiae, Mycobacterium, Pseudomonas, Legionella, Burkholderia, by genomic study. Have you seen them on the bacterial culture of any other methods to verify your results as well?

10. Add bootstrapping value in the phylogenetic tree.

Reviewer #2 (Public repository details (Required)):

Genomes need to be uploaded to Genbank

Reviewer #2 (Comments for the Author):

See attached word document, thank you

Staff Comments:

Preparing Revision Guidelines

Please return the manuscript within 60 days; if you cannot complete the modification within this time period, please contact me. If you do not wish to modify the manuscript and prefer to submit it to another journal, please notify me of your decision immediately so that the manuscript may be formally withdrawn from consideration by Microbiology Spectrum.

The research article by Gu *et al.* examined seven clinically proven pathogenic strains of *acanthamoeba* and compared them with those of *acanthamoeba* and other amoebae from public databases to study genomic, phylogenetic, and identification of virulence genes specific to *acanthamoeba*. The authors also demonstrated rhizome gene mosaicism analyses to explore amoeba symbiont interactions that possibly contribute to pathogenesis. Though the manuscript is interesting, well-written, and easy to read and understand, I still have some suggestions/ concerns that need to be addressed before this manuscript is published. My suggestions/concerns are listed below:

1. Line 65-66: To date, 23 genotypes of *acanthamoeba* are known, i.e., T1-T23. According to a recent study, *acanthamoeba* has been isolated from public freshwater sources in Thailand and named as *A. bangkokensis*, and it belongs to the T23 genotype. (doi: 10.1038/s41598-021-96690-0) that need to be updated in the introduction section.
2. Line 82-83 what does mean “selective grazing growth.” As per my understanding, it reflects phagocytosis. So my question is phagocytosis in *acanthamoeba* specific or non-specific?
3. Line 85-86: How can *acanthamoeba* resist phagocytosis? It is reported that endosymbiont survives intracellularly inside *acanthamoeba* and resists acanthamoebacidal digestion. However, I can’t understand how *acanthamoeba* can resist phagocytosis.
4. Italic *acanthamoeba* and all bacteria names throughout the text.
5. Line 121 authors mentioned that they had studied the “prevalence of endosymbionts.” However, results are nowhere mentioned and discussed, and seven is a minimal sample size to determine prevalence.
6. In the material and method, it is mentioned that to culture *acanthamoeba*, non-nutrient agar method was used, i.e., monoxenic culture, which includes cultivation of *acanthamoeba* in the presence of bacteria such as *E.coli*. How about axenic culture .i.e. without any bacteria? For all genomic studies, DNA extraction should be done with pure culture. DNA extracted from monoxenic culture probably has *E.coli* DNA as well. Justify it
7. Line 353: write the full form of ARB and ARVs
8. A horizontal gene transfer study was done with new and publicly available data. Have authors tried to validate the same with in *vitro* experiments
9. Line 475-477 authors have identified several endosymbionts, such as *Chlamydiae*, *Mycobacterium*, *Pseudomonas*, *Legionella*, *Burkholderia*, by genomic study. Have you seen them on the bacterial culture of any other methods to verify your results as well?
10. Add bootstrapping value in the phylogenetic tree.

POINT-BY-POINT RESPONSES TO REVIEWERS' COMMENTS

Reviewer #1

The research article by Gu et al. examined seven clinically proven pathogenic strains of acanthamoeba and compared them with those of acanthamoeba and other amoebae from public databases to study genomic, phylogenetic, and identification of virulence genes specific to acanthamoeba. The authors also demonstrated rhizome gene mosaicism analyses to explore amoeba symbiont interactions that possibly contribute to pathogenesis. Though the manuscript is interesting, well-written, and easy to read and understand, I still have some suggestions/ concerns that need to be addressed before this manuscript is published.

Response: We sincerely appreciate all your positive comments. The suggestions/concerns were addressed in detail as below.

- 1. Line 65-66: To date, 23 genotypes of acanthamoeba are known, i.e., T1-T23. According to a recent study, acanthamoeba has been isolated from public freshwater sources in Thailand and named as A. bangkokensis, and it belongs to the T23 genotype. (doi: 10.1038/s41598-021-96690-0) that need to be updated in the introduction section.*

Response: Thank you so much for pointing this out. We have updated the introduction section accordingly. Please refer to correspondingly text on **Line 65-66** in our revised manuscript.

- 2. Line 82-83 what does mean "selective grazing growth." As per my understanding, it reflects phagocytosis. So my question is phagocytosis in acanthamoeba specific or non-specific?*

Response: Thank you so much for your question. We should have described it much clearer. We have briefly mentioned in our manuscript that *Acanthamoeba* feed on bacteria, fungi, yeasts, algae during the trophozoite stage. Protozoan grazing is believed to be the major trophic pathway whereby the biomass production and amoebae is indicated to a role as bacterial regulators which help to drive bacterial virulence and may contribute to the spread of antibiotic resistance. The amoebae can exert a pressure of selection on the microorganisms

that live in their environment and select the microorganisms able to live inside cells in any situations. In fact, any object larger than 0.5 μm can be engulfed, including latex beads. Therefore, the phagocytosis in acanthamoeba is non-specific which is not needed for recognizing specific markers on microorganisms to ingest them. And the phagocytosis of microorganisms follows the endocytic pathway to be degraded in acidic phagolysosomes by a number of hydrolases. In our revised manuscript, we have modified the corresponding part to make it more understandable.

3. *Line 85-86: How can acanthamoeba resist phagocytosis? It is reported that endosymbiont survives intracellularly inside acanthamoeba and resists acanthamoebacidal digestion. However, I can't understand how acanthamoeba can resist phagocytosis.*

Response: Thank you for your valuable comment. To clarify this point more clearly in our study, we have inserted the detailed explanations of the relationships of amoeba-resisting microorganisms (ARMs), amoeba-resistant bacteria (ARB) as well as endosymbionts. Amoebae can phagocytose and be infected by several microorganisms at the same time, and they have variable relationships with these microorganisms. In some cases, the microorganism becomes prey for the amoebae, which feed on surrounding microorganisms, in particular, bacteria. In other cases, the introduced microorganisms are parasitic and pathogenic for the amoeba and can destroy their host. In other instances, the amoebae can maintain symbiotic relationships with introduced microorganisms. The corresponding changes have been made on **Line 85-93** of the revised manuscript.

4. *Italic acanthamoeba and all bacteria names throughout the text.*

Response: We thank the suggestion and have done so accordingly. **We have thoroughly revised the writing in our revised manuscript.**

5. *Line 121 authors mentioned that they had studied the "prevalence of endosymbionts." However, results are nowhere mentioned and discussed, and seven is a minimal sample size to determine prevalence.*

Response: Thank you for your valuable comment. We agree with the reviewer on the issue that seven sample size is not enough to evaluate the prevalence of endosymbionts. It will be much convincing that if more clinical samples were included in the analysis. However, the limitations analytical methods and the large sample data increase the difficulty of analysis. In our revised manuscript, we have replaced the incorrectly used word "prevalence" and discussed this limitation in the Discussion section. Please refer to the correspondingly text on **Line 127-128** and **Line 421-423 (Discussion section)** in our revised manuscript.

6. *In the material and method, it is mentioned that to culture acanthamoeba, non-nutrient agar method was used, i.e., monoxenic culture, which includes cultivation of acanthamoeba in the presence of bacteria such as E.coli. How about axenic culture .i.e. without any bacteria? For all genomic studies, DNA extraction should be done with pure culture. DNA extracted from monoxenic culture probably has E.coli DNA as well. Justify it*

Response: We sincerely appreciate your valuable comments. We understand your concern and your suggestion really means a lot to us. We have briefly described the cultivation method in the method part that *Acanthamoeba* are grown on agar-agar plates seeded with *Escherichia coli* (ATCC25922) and Page's modified Neff's amoeba saline. To further identify virulence factors and endosymbionts in *Acanthamoeba*, we removed scaffolds that originated from *E.coli*. On the one hand, *E.coli* becomes prey for the the amoebae among all relationships while we aim to find pathogenic endosymbionts in *Acanthamoeba*. On the other hand, the pathogenic effect of *E.coli* remains largely unrevealed, and whether it has a pathogenic effect remains to be explored. Axenic culture is a very good option to study *Acanthamoeba* without interference, and we have started to adopte your suggestion and apply it to the next research. Besides, the role of *E.coli* in AK will also become our future research direction.

7. *Line 353: write the full form of ARB and ARVs*

Response: We thank the reviewer for point this out. We feel really sorry for our carelessness. **We have rectified the full form of ARB and ARVs in our revised manuscript.**

8. *A horizontal gene transfer study was done with new and publicly available data. Have authors tried to validate the same with in vitro experiments.*

Response: Thank you for your valuable comment. We're sorry that we cannot timely present the vitro experiments based on current research findings. At present, we have started research on experiments verification, and in the next research, we will explore these interesting results with more evidence.

9. *Line 475-477 authors have identified several endosymbionts, such as Chlamydiae, Mycobacterium, Pseudomonas, Legionella, Burkholderia, by genomic study. Have you seen them on the bacterial culture of any other methods to verify your results as well?*

Response: Thank you very much for your nice comments. We feel sorry that we did not provide enough information about endosymbionts in our clinical isolates. However, several endosymbionts have been verified in similar clinical isolates, including *Legionella*, *Pseudomonas*, *Mycobacterium* and *Chlamydothila* family with vitro experiments. The paper may help you to understand and the experimental methods are worth learning in our next studies. (doi: 10.1016/j.optha.2009.08.033)

10. *Add bootstrapping value in the phylogenetic tree.*

Response: Thank you so much for your suggestion. We have also added the changes in the new phylogenetic tree of our revised manuscript. Besides, we added published T3 and T4 subtypes and alleles to our phylogenetic tree to support our claim that there is a diversity of isolates. The Results (page 7), Discussion (page 16-17) and Material sections (page 22) have also been updated.

Reviewer #2

This seems to be two papers squeezed into one. A comparative genomes paper, comparing the different clinical isolates to published genomes would likely have been enough for one

paper. By adding in the HGT, you are squeezing a lot of information into one paper, but not really giving either section its due. Your aggressive criteria for identifying AK-related genes likely left you with too small of a pool. I would expect you would have some controls to ensure you still captured known genes like mannose-binding- protein but they likely got discarded based on your criteria. An absence/presence criteria for genes seems to have not worked in your favor, when it is likely that it is variability in the genetic sequence and amino acids that lend to pathogenicity.

Likewise, the HGT work is interesting but you never related it back to known pathways for those bacteria to cause microbial keratitis. Example: Tupanviruses are known to sequence Mannose-binding protein; so why did you not look for known pathogenic genes from bacteria in Acanthamoeba if that was the objective (to find genes that may lend to pathogenicity). May want to consider if this is one paper or two, and if you keep it as one, focus the message more since it definitely gets lost under all of the extra information.

Response: We sincerely appreciate all your positive comments. We understand your concern. Please also find the following changes we have made, which were detailed below.

Row 47, blinding, not blinging

Row 48, phenomenon

Row 49: underlying opportunistic pathogenic mechanism, maybe

Row 78, blindness

Row 136 Naegleria

Row 295 weak

Response: Thanks for your careful checks. We feel really sorry for the improper wordings.

Rows 71-74, split sentence

Response: Thanks for your help. Based on your comments, we have rearranged and rewrote sentences on **page 4** in the revised manuscript to make it more comprehensible.

Row 79 Unsure what “shortening of clinical course” means

Response: We thank the reviewer for point this out. We should have described it much clearer. Acanthamoeba keratitis is often misdiagnosed and treated as herpetic, bacterial, or mycotic keratitis, as many signs and symptoms may look similar to other kinds of keratitis. It is challenging for an ophthalmologist to find the right diagnosis. Therefore, diagnosis is often delayed and ophthalmologists tend to observe a heterogeneous and protracted clinical course. We have corrected the formulation on page 4 in the revised manuscript.

Row 83; do you mean “grow” here? Or remove growth entirely, also makes sense

Response: Thanks for your suggestions. We have corrected it according to your suggestion, yes, it would be more understandable.

1. Methods comment: Please explain how you selected for Acanthamoeba’s whole genome during sequencing versus any internal endosymbionts. Results indicated that endosymbionts were present in many clinical isolates but that was not clear if that was based on gene results or actual identification of endosymbionts within the Acanthamoeba. If endosymbionts were found within clinical isolates, please describe how they were identified.

Response: Thank you for your valuable comment. We should have described it much clearer. We have briefly described the methods in our manuscript that the predicted proteins were searched for in the NCBI GenBank protein sequence database (nr) and Cluster of Orthologous Group of proteins (COG) database via BLASTp program. Then we identified the sequences homologous of the predicted proteins best matching with proteins from amoeba-resistant microorganisms. From our current study, the endosymbionts results were based on gene results. However, many papers have proved through in vitro experiments that there are indeed many endosymbionts in *Acanthamoeba* and we will explore the interesting results with more evidence in the next research. This paper may help you to understand. (doi: 10.1016/j.ophtha.2009.08.033)

2. Rows 176-189. As some of the non-Acanthamoeba that were compared to are

“pathogenic”, ie Naegleria, aren’t you concerned you would be discarding potential hits by removing anything designated as non-pathogenic. Similar pathways between organisms seem likely to cause infection. The limitations placed here, especially around AK vs non-AK isolates may have limited your results. Genetic differences in genes are just as likely to play a role in pathogenesis vs full on gene presence or absence between isolates. It seems like the paper had limited hits because of this overly selective method of analysis.

Response: Thank you for this important point. We understand your concern and your suggestion really means a lot to us. As a matter of fact, there exists no readymade virulence gene bank for Acanthamoeba in the Virulence Factor Database (VFDB). There are still many limitations in the mining of virulence genes and the pathogenesis mechanism of AK remains unknown. In our study, our choose of comparative genomic analysis to find virulence genes lay the groundwork for further digging into pathogenesis in the future. We will gradually complete the Acanthamoeba virulence gene pool in the next study.

3. *Row 228-235; missing reference to figure 1. Recommend adding non-Acanthamoeba amoeba to phylogenetic tree for scale. This paper (<https://pubmed.ncbi.nlm.nih.gov/32630775/>) may help with your discussion around the variable pathogenicity of T4 genotypes. It does look like you have a variety, may want to add additional published strains to the tree to help support this claim.*

Response: We sincerely appreciate the reviewer for this helpful suggestion and corrected the image as suggested. We have also replaced the original figures with the **new image of phylogenetic tree accordingly**, please refer to figures accordingly in our revised version. And we have also updated the **Results (page 7), Discussion (page 16-17) and Material sections (page 22)**.

4. *Row 285, again some of those amoeba do cause human disease, if not keratitis. Selecting only genes in Acanthamoeba that cause AK vs pathogenic but non-AK amoeba may remove highly critical genes in pathogenesis that are necessary steps in causing disease. I do not see any discussion of mannose binding protein in your paper, which is*

considered a critical gene for Acanthamoeba keratitis pathogenesis.

Response: Thank you for this important point. We have identified orthologous genes using Proteinortho with some controls. To minimize highly critical genes loss, we lowered the control limits for coverage and identity and still did not find the mannose binding protein gene. The results may indicated that genomic insufficiency or similar infection pathways in different *Acanthamoeba*, which were needed to be clarified in the future. We have also modified the discussion of this conclusion, please refer to part accordingly in our revised version.

5. *Row 339, you state you found Pseudomonas in 5/ of your isolates. Is this based on gene sequencing ((through whole genomes, where you found Pseudomonas genes,) or gene sequencing where Pseudomonas was found to be a true endosymbiont of that isolate? This is not clear in the whole section and makes it very confusing.*

Response: We sincerely appreciate your valuable comments. Our results were based on whole genomic sequencing from which we found predicted proteins sequences of isolates best matching with *Pseudomonas*. We feel sorry for our poor writings. For your convenience and provide the reader with a more intuitive conclusion, we have made extensive modifications to corresponding conclusion section and added necessary Supplementary Material Table S2 data to evidence our results.

6. *Rows 335-338: Pseudomonas can definitely be cultured outside their host, this sentence is very confusing. In general, it is hardly surprising to see Pseudomonas in Acanthamoeba or their genes, they are equally ubiquitous as Acanthamoeba. You may want to draw a stronger tie to microbial keratitis caused by Pseudomonas (did you find any Pseudomonas pathogenesis related genes in Acanthamoeba for example since that pathway is well established?) than just saying it was very prevalent in Acanthamoeba.*

Response: We feel great thanks for your professional review work on our article. The stable associations of bacteria endosymbionts with amoebae leading to long term symbiotic

interactions. None of these bacterial endosymbionts have the ability to survive and cannot be cultured outside their amoebic host cells because of *Acanthamoeba* might be able to protect them. According to your advice, we have revised the corresponding texts in our manuscript. Moreover, your suggestion really means a lot to us. Though previous studies have focused on the role of *Legionella pneumophila* in AK which has been the most studied model, the specific underlying pathogenic mechanism remains largely unrevealed. Our work may provide with new insights into pathogenesis with a focus on *Pseudomonas*, thus our study is innovative and promising. Comparing to *L. pneumophila* effector proteins can support intracellular growth in the *Legionella*-containing vacuoles (LCV) and manipulate host cell death, *P. aeruginosa* rapidly colonized and killed biofilm-associated amoebae using type III secretion system (T3SS) also have similarity. Thus, we hope to further study the role of *Pseudomonas* in the pathogenesis of AK with differential expression of virulence genes in endosymbiotic *Pseudomonas aeruginosa* in the next study. These studies may help you to understand.

(doi: 10.1038/ismej.2008.47)

(doi: 10.1080/21505594.2017.1373925)

7. *Row 373: Pseudomonas genes? Or did you actually find bacteria in your clinical isolates?*

Response: We thank the reviewer for point this out. We have revised the corresponding conclusion section.

8. *Row 400-401: rephrase, half of sentence is redundant*

Response: We sincerely appreciate this helpful suggestion. We have rephrased this sentence on Line 317-318 as you suggested.

9. *Row 431-434: Citation needed for this statement.*

Response: Thank you for pointing this out. We have inserted and updated the citation for this statement, please refer to the correspondingly text on **Line 352-355** in our revised manuscript.

10. Row 478-480: How can Pseudomonas be overrepresented in the genotype T3 when you only had one isolate to compare to 4 isolates of T4?

Response: We sincerely appreciate your valuable comments. We agree with the reviewer on the issue that one isolate belonging to genotype T3 is not enough to evaluate the the overexpression of Pseudomonas. It will be much convincible that if more clinical samples were included in the analysis. **We have revised this conclusion in the revised manuscript** and hope to confirm this in subsequent experiments.

11. Row 528-529 Not sure the study supports this statement

Response: We sincerely appreciate your valuable comments. We accept your suggestion that this study does not currently demonstrate this, we have modified the contents accordingly on **page 21** of our revised manuscript and hope to confirm this in the next experiment.

12. Row 485-488: This paper in no way supports your study. This was showing the physical presence of Pseudomonas helps Acanthamoeba colonize a lens not anything related to HGT. . I would remove.

Response: Thank you for this important point. We agree with the reviewer on the issue that this study shows *Pseudomonas* can actually cooperate in the pathogenesis of contact lens related AK: it increases the resistance of Acanthamoeba to contact lens disinfectants and colonization on the contact lens. Based on your comments, **we have made the corrections to make the unit harmonized within the whole manuscript.**

March 22, 2022

Dr. Meiqin Zheng
Wenzhou Medical University
Chashan University Town
Wenzhou
China

Re: Spectrum00025-22R1 (A comparative genomic approach to determine the virulence factors and horizontal gene transfer events of clinical *Acanthamoeba* isolates.)

Dear Dr. Meiqin Zheng:

Thanks for addressing the reviewers' constructive input.

Your manuscript has been accepted, and I am forwarding it to the ASM Journals Department for publication. You will be notified when your proofs are ready to be viewed.

Sincerely,

Tim Downing
Editor, Microbiology Spectrum

Journals Department
Supplemental Dataset: Accept
Supplemental Dataset: Accept
Supplemental Dataset: Accept